# Growth Hormone Upregulates Melanocyte-Inducing Transcription Factor Expression and Activity via JAK2-STAT5 and SRC Signaling in GH Receptor-Positive Human Melanoma

**DOI:** 10.3390/cancers11091352

**Published:** 2019-09-12

**Authors:** Reetobrata Basu, Prateek Kulkarni, Yanrong Qian, Christopher Walsh, Pranay Arora, Emily Davis, Silvana Duran-Ortiz, Kevin Funk, Diego Ibarra, Colin Kruse, Samuel Mathes, Todd McHugh, Alison Brittain, Darlene E. Berryman, Edward O. List, Shigeru Okada, John J. Kopchick

**Affiliations:** 1Edison Biotechnology Institute, Ohio University, Athens, OH 45701, USA; basu@ohio.edu (R.B.); pk585316@ohio.edu (P.K.); qiany@ohio.edu (Y.Q.); ed218514@ohio.edu (E.D.); sd504111@ohio.edu (S.D.-O.); funkk@ohio.edu (K.F.); di832513@ohio.edu (D.I.); ck178807@ohio.edu (C.K.); mathes@ohio.edu (S.M.); tm912015@ohio.edu (T.M.); ab647713@ohio.edu (A.B.); berrymad@ohio.edu (D.E.B.); list@ohio.edu (E.O.L.); okada@ohio.edu (S.O.); 2Molecular and Cellular Biology (MCB) Program, Department of Biological Sciences, Ohio University, Athens, OH 45701, USA; 3Department of Biomedical Sciences, Heritage College of Osteopathic Medicine, Ohio University, Athens, OH 45701, USA; cw763712@ohio.edu (C.W.); pa727717@ohio.edu (P.A.); 4Department of Chemistry and Biochemistry, Ohio University, Athens, OH 45701, USA; 5The Diabetes Institute, Ohio University, Athens, OH 45701, USA; 6Department of Pediatrics, Heritage College of Osteopathic Medicine, Ohio University, Athens, OH 45701, USA

**Keywords:** growth hormone receptor (GHR), growth hormone (GH), melanoma, JAK, STAT, SRC (sarcoma or c-Src), chemoresistance, MITF

## Abstract

Growth hormone (GH) facilitates therapy resistance in the cancers of breast, colon, endometrium, and melanoma. The GH-stimulated pathways responsible for this resistance were identified as suppression of apoptosis, induction of epithelial-to-mesenchymal transition (EMT), and upregulated drug efflux by increased expression of ATP-binding cassette containing multidrug efflux pumps (ABC-transporters). In extremely drug-resistant melanoma, ABC-transporters have also been reported to mediate drug sequestration in intracellular melanosomes, thereby reducing drug efficacy. Melanocyte-inducing transcription factor (MITF) is the master regulator of melanocyte and melanoma cell fate as well as the melanosomal machinery. MITF targets such as the oncogene MET, as well as MITF-mediated processes such as resistance to radiation therapy, are both known to be upregulated by GH. Therefore, we chose to query the direct effects of GH on MITF expression and activity towards conferring chemoresistance in melanoma. Here, we demonstrate that GH significantly upregulates MITF as well as the MITF target genes following treatment with multiple anticancer drug treatments such as chemotherapy, BRAF-inhibitors, as well as tyrosine-kinase inhibitors. GH action also upregulated MITF-regulated processes such as melanogenesis and tyrosinase activity. Significant elevation in MITF and MITF target gene expression was also observed in mouse B16F10 melanoma cells and xenografts in bovine GH transgenic (bGH) mice compared to wild-type littermates. Through pathway inhibitor analysis we identified that both the JAK2-STAT5 and SRC activities were critical for the observed effects. Additionally, a retrospective analysis of gene expression data from GTEx, NCI60, CCLE, and TCGA databases corroborated our observed correlation of MITF function and GH action. Therefore, we present in vitro, in vivo, and in silico evidence which strongly implicates the GH–GHR axis in inducing chemoresistance in human melanoma by driving MITF-regulated and ABC-transporter-mediated drug clearance pathways.

## 1. Introduction

The current estimations of the American Academy of Dermatology (AAD) state that more than 1 million Americans are currently suffering from melanoma with >100,000 new cases projected in 2019 (as of 16th June, 2019 at www.aad.org/media/stats/conditions/skin-cancer). AAD estimates show 20 melanoma deaths/day in USA as well as an 800% increase in melanoma incidence in female patients (18–39 years) between 1970–2009. The National Cancer Institute’s (NCI) Surveillance, Epidemiology and End Results (SEER) Program corroborate the figures. By stage, the five-year survival rates between 2009–2016 at NCI’s database for localized, regional metastasis, and distant metastasis melanoma, goes from 99% to 65% to 25%, respectively (as of 16th June, 2019 at www.seer.cancer.gov/statfacts/html/melan.html), in spite of the current advances in therapeutic options and efficacy. Although they constitute <2% of all incident cancers, melanoma is the fifth most lethal due to an intrinsic resistance to standard chemotherapy [1], radiation therapy [2], targeted therapies such as BRAF inhibitors [3,4], and immunotherapy such as immune checkpoint inhibitors (ICI) [5]. Therefore, melanoma is an excellent model of studying the mechanisms underlying therapy refractoriness in human cancers. Incidentally, melanoma cells were also found to have the highest level of growth hormone receptor (GHR) expression among the 60 different cancer cell lines in the NCI-60 panel [6].

The growth hormone (GH) and growth hormone receptor (GHR) ligand–receptor pair are expressed in almost all types of human cancers and have been identified as a validated drug target by in vitro or in vivo studies in more than twelve cancer types [7]. Circulating GH is produced centrally by the pituitary somatotrophs and peripherally by several tissues including skin [8,9]. GH regulates longitudinal growth, organ development, as well as carbohydrate, protein, and lipid metabolism in a tissue- and sex-specific manner [8]. GH action is mediated in several tissues directly, as well as indirectly by its secondary effector insulin-like growth factor 1 (IGF1). In normal skin, fibroblasts and melanocytes harbor a significantly higher GHR and IGF1 transcript expression compared to the keratinocytes, whereas the keratinocytes are enriched with IGF1 receptors (IGF1R) and function in maintaining epidermal homeostasis [9,10]. Increased GH responsiveness was identified by immunohistochemical studies targeting GHR on melanoma patient-derived paraffin samples [11]. A large body of research in the last 20 years has implicated GH action in the regulation of multiple oncogenic processes such as proliferation, migration, invasion, anchorage-independent growth, as well as resistance to chemo- and targeted therapy in human tumors [12,13,14]. However, the essential molecular underpinnings of these GH-regulated oncogenic processes are only recently coming to light. In colon cancer, DNA damage-induced GH production was found to suppress p53-mediated apoptosis [15,16], while autocrine/paracrine GH also attenuated efficacy of irradiation and mitomycin-C or doxycycline treatment in breast and endometrial cancers by suppressing apoptosis [17,18]. In GHR-expressing (GHR+) cancers of breast, pancreas, colon, liver, endometrium, as well as melanoma, GH induces the process of epithelial-to-mesenchymal transition [19,20], which is a critical step in therapy evasion and invasive metastasis in tumors. Further, we and others have shown a direct effect of GH–GHR action in upregulating expression of ATP-cassette-containing (ABC) multidrug efflux pumps such as ABCB1, ABCC1, ABCC2, and ABCG2 in melanoma [21] as well as in breast cancer [22]. Increased GH production in other tissues, such as lungs, has been found to guide lung metastasis of B16F10 melanoma xenografts in DJ1-KO mice [23]. Therefore, it appears that the GH–GHR pair is harnessed to particularly evade therapeutic challenges by advanced melanoma tumors.

Drug sequestration in intracellular compartments such as lysosomes and endosomes dampens the cytotoxic effects of anticancer drugs [24,25]. Melanoma tumors are uniquely enabled to sequester anticancer compounds into intracellular vesicles called melanosomes, which is also the site of melanin synthesis. A series of elegant studies by Chen et al. exhibited the melanosomal sequestration of cisplatin and increased tyrosinase activity following increasing doses of cisplatin treatment in both melanotic MNT1 and amelanotic SK-MEL-28 cells [26,27,28]. Both cells exhibited increase in stage-II melanosomal marker gp100/PMEL17 coinciding with increased drug resistance [26,27,28]. Chen et al. further proposed the ‘ABC-M model’ detailing the role of ABC-transporters like ABCB1, ABCB5, ABCB8, ABCC1, ABCC2, and ABCG2 in active drug efflux away from the cytoplasm—either outside the cell or into melanosomal compartments [28]. In fact, treatment of melanoma with multiple anticancer compounds such as dacarbazine and vemurafenib also leads to selection towards ABCB5-enriched tumors [29,30,31,32,33]. Both melanosome synthesis and melanogenesis pathway are regulated in melanoma by the melanocyte-inducing transcription factor/micropthalmia-associated transcription factor (MITF). In 2005, MITF was identified as a ‘lineage survival oncogene’ in human melanoma, with significant amplification in metastatic stage and correlated with decreased overall survival in melanoma patients [34,35]. It was previously shown that GH action in GHR+ human melanoma cells upregulates while GHR knockdown suppresses the ABC-transporter system, especially ABCB1, ABCB5, ABCB8, ABCC1, ABCC2, ABCG1, and ABCG2 differentially in a drug-dependent manner [21]. Mouse B16F10 tumor xenografts in bovine GH transgenic (bGH) mice also exhibited a significantly upregulated basal (without drug treatment) expression of the ABC-transporters compared to the same in wild-type (WT) littermates (Qian et al. personal communications). In addition, it is reported that alpha-melanocyte stimulating hormone (α-MSH) and MITF promote the expression of MITF-target and oncoprotein MET allowing MET-ligand HGF-mediated antiapoptotic effects [36]. In relevance, the GHRKO and the Ames’ mice have lower α-MSH immunoreactivity in the hypothalamus [37]; while a GH-regulated HGF-MET expression was observed in human melanoma cells [20]. Further, GHR and MITF clustered together among the top 20 differentially upregulated genes between human melanoma (metastatic and primary) sample gene expression data extracted from the GSEA7553 set and ranked by high and low PGC1A expression levels [38]. In the current study, here we therefore hypothesized that the GH–GHR system could be involved in upregulating the process of melanosomal drug sequestration actively through MITF or passively via ABC-transporters, or both, and thereby contributing to the observed GH-mediated drug resistance. To test this hypothesis, we used multiple human melanoma cells—SK-MEL-30, MDA-MB-435, SK-MEL-28, and MALME-3M, and we found that in both melanotic and amelanotic melanoma, the GH–GHR system upregulates both ABC-transporter levels as well as MITF-directed processes of melanosome formation and tyrosinase activity, differentially, following multiple chemotherapy treatments. We verified our findings in vivo in B16F10 tumor xenografts in bGH vs. WT mice. Additionally, to validate our experimental observations, we performed a retrospective analysis of GHR expression and MITF and MITF-targets using the RNA-seq data from multiple publicly available cancer cell line datasets (NCI60 and CCLE), normal human skin RNA expression in the GTEx dataset, as well as of human melanoma patients from the Cancer Genome Atlas (TCGA) database.

## 2. Results

### 2.1. Autocrine GH Production Correlates with Upregulated MITF Expression Following Drug Treatment in Human Melanoma Cells

In human cancers, autocrine ligand–receptor loops are known to be highly oncogenic. Autocrine GH in GHR-expressing breast cancers have been shown to possess higher metastatic and invasive potential [12,38,39,40,41]. In our previous study, we observed presence of GH1 (hereafter called GH) RNA expression in melanoma cell lysates [20,21]. In mammary and endometrial cancers, autocrine GH provides protection from radiation therapy [17]. Therefore, we closely monitored the production of GH and GHR RNA production in human melanoma cells following addition of anticancer drugs, across four different time points. We used four different drugs—varied by their mechanism of action—on SK-MEL-28 cells. In all cases, SK-MEL-28 showed a significant increase either in GH or GHR or both within 12 h of drug addition (Figure 1A–D). Doxorubicin and crizotinib elicited a 1.5–2-fold increase in GH RNA, while crizotinib, cabozantinib, and vemurafenib elicited more than 2-fold increases in GHR levels within 24 h of drug addition (Figure 1A–D). Concomitantly, we observed 1.5-fold to >2-fold increase in MITF RNA levels within 24 h for all the four drugs (Figure 1A–D). A similar inquiry into the MALME-3M and MDA-MB-435 human melanoma cells only at the 24 h timepoint following drug addition revealed similar and even more significant increases in GH and MITF levels (Figure 1E–H). In response to cisplatin, vemurafenib, and doxorubicin treatments MDA-MB-435 cells showed a 22-, 21-, and 10-fold increases, respectively, in GH RNA levels (Figure 1E) while the MITF levels increased by 1.2–1.7-fold at 24 h (Figure 1G). MALME-3M cells also showed a concurrent significant rise in GH and MITF levels at 24 h (Figure 1F,H). The GHR transcript levels did not change significantly at the 24 h timepoint in MDA-MB-435 and MALME-3M cells. Interestingly, the ABC-transporters, which are involved in multidrug efflux out of the cell as well as active loading of anticancer drugs into melanosomal compartments [28], were also increased with the increase in GH and MITF. Corroborating previous reports by us and others [21,22], the ABC-transporter levels were also upregulated at the 24 h timepoint after drug addition in a context dependent manner. In SK-MEL-28 cells, ABCG1 was upregulated 2–10-fold in response to doxorubicin, vemurafenib, and cabozantinib; ABCB1 was upregulated 3–10-fold in response to doxorubicin and crizotinib; while crizotinib also elicited a seven-fold increase in ABCC4 and almost four-fold increase in ABCB8 (Appendix A). A dose dependent effect of GH treatment on ABCB1, ABCC1, and ABCG2 was observed also at protein level, further bolstering our observation (Appendix A). We also observed a 1.5- to more than two-fold increase the GH protein levels in MALME-3M, SK-MEL-28, and MDA-MB-435 cell lysates 48 h after drug addition (Figure 1J,K). The MITF protein levels were significantly different at the 48 h timepoint in MALME-3M cells (Figure 1J,L). These results indicate that GH action and MITF might be playing a role in conjunction to overcome the effects of therapeutic challenges in melanoma cells. We next wanted to identify if GH directly causes an upregulation in MITF expression and activity in melanoma.

### 2.2. GH Action Directly Modulates MITF and MITF Target Gene Expression Levels in Human Melanoma Cells

MITF plays a central role in regulating melanocytic lineage as well as phenotype switching in melanoma [34,35] and elicits a tightly regulated survival advantage to melanoma [42], especially under therapeutic challenge [43]. Therefore, to better understand the effect of GH on MITF expression, we treated human melanoma cells with increasing doses (0, 50, 100, 200 ng/mL) in presence or absence of drug and followed the expression of MITF and MITF target genes across three time points (6, 24, and 48 h) in both the melanotic melanoma cell SK-MEL-30 and the amelanotic cell SK-MEL-28. A variation in MITF expression can be orthogonally confirmed by analyzing the expression level of its target genes involved in multiple oncogenic processes such as melanogenesis (TYR, TYRP1, DCT/TYRP2, MELANA, PMEL), cell survival regulation (BCL2, HIF1A), cell cycle regulation (CDKN1A, CDKN2A), cell invasion (MET, SLUG/SNAI2), and mitochondrial metabolism (PPARGC1A/PGC1A). The chosen targets are widely known and studied as established MITF targets [44,45].

We observed a differential response to GH in these two cell lines. In the amelanotic cell line SK-MEL-28, exogenous GH itself did not elicit a significant upregulation of MITF and MITF-targeted melanogenesis genes irrespective of doxorubicin; although TYRP1 and MLANA had a modest 1.5-fold increase at higher concentrations of GH only at 6 h in presence of doxorubicin (Figure 2A,B). However, a number of the MITF-targets not involved in melanogenesis were significantly upregulated by GH-excess in SK-MEL-28: (i) PGC1A, HIF1A, PMEL, BCL2, and CDKN1A were upregulated by 1.5–2-fold at different time-points in absence of GH, while (ii) doxorubicin addition saw a similar significant increase in PGC1A, BCL2, BRCA1, DCT/TYRP2, and CDKN1A at different time-points. The ABC-transporters ABCB8 and ABCG2 were consistently upregulated by GH addition even in absence of doxorubicin, whereas addition of doxorubicin markedly enhanced the scale and length of ABCB8 and ABCG2 upregulation (Figure 2A,B). SK-MEL-28 cells transfected with GHR-siRNA showed a marked suppression of MITF and MITF-targets TYRP1 and MET compared to scramble-siRNA transfected samples in both presence and absence of additional GH (Figure 2C–E). In the melanotic cell line SK-MEL-30, in absence of doxorubicin, GH only induced a transient (detected mostly at 24 h timepoint) and mild elevation in MITF levels, producing a 1.5–2-fold increase only at higher concentrations of GH for the MITF-target genes MLANA, ABCB5, BCL2, BRCA1, and CDKN1A (Appendix A). However, in presence of doxorubicin, firstly, there was an upregulation in the GH–GHR axis (as observed in the previous section) with a 1.7-fold increase in GH at 24 h and >2-fold increase in GHR levels at both 24 and 48 h timepoints, at conditions of GH excess (Appendix A). For MITF, a more acute (detected at 6 h timepoint) 1.7-fold upregulation was observed in 6 h with a sustained (detected at 48 h timepoint) expression of its target genes (Appendix A). Genes involved in the melanogenesis pathway such as TYR, TYRP1, MLANA, as well as the melanosomal marker PMEL were acutely upregulated at the 6 h timepoint. In SK-MEL-30 in the presence of doxorubicin, other known MITF targets such as BRCA1, BCL2, and CDKN1A were also significantly upregulated at 24 h while the ABC-transporter ABCB5, implicated in loading anticancer drugs into melanosomes [28], was significantly upregulated by two-fold at 6, 24, and 48 h timepoints specifically at conditions of GH-excess (Appendix A).

We further verified our results using the melanotic cell lines MALME-3M and MDA-MB-435 only at the 24 h timepoint, using doxorubicin, vemurafenib, and cisplatin as drug treatments. For both cell lines, the maximum response was obtained with vemurafenib treatment (Appendix A). In MDA-MB-435, in addition to upregulating MITF levels by 1.5-fold, vemurafenib treatment also significantly elevated the levels of melanogenic MITF-targets genes PMEL, TYR, TYRP1, DCT, MLANA, as well as PGC1A, and MET (Appendix A). Additionally, all the drugs sharply increased HIF1A in MDA-MB-435 (Appendix A). In MALME-3M cells, vemurafenib treatment led to a two-fold increase in MITF RNA levels (Appendix A), as well as significant upregulation in MITF-targets TYRP1 (>20-fold), DCT/TYRP2 (>eight-fold), PMEL, and MLANA, as well as PGC1A and MET (Appendix A). Therefore, a sharp increase in the melanogenic pathway was observed following drug treatment. Addition of 50 ng/mL GH, however, did not display a clear increase in the above gene expression in either cell lines. siRNA-mediated GHRKD significantly suppressed the drug-induced elevation in MITF and MITF target genes (Appendix A), indicating the role of high levels of autocrine GH production due to drug treatment as shown in the earlier section. The levels of endogenous IGF1, PRL, and PRLR did not vary by more than 1.5-fold following drug addition or GHRKD in MDA-MB-435 or MALME-3M (not shown here). Additionally, we treated mouse melanoma cell line B16F10 with 0, 50, and 500 ng/mL of bovine GH (bGH) for 24 and 48 h and analyzed the change in expression of MITF and a number of MITF target genes. We observed a robust effect of 500 ng/mL bGH, which consistently upregulated the expression of MITF as well as its target genes *Tyrp1*, *Dct*, *Mlana*, *Hif1a*, and *Bcl2* significantly, especially at the 48 h time point (Appendix A). These detailed analyses demonstrate a distinct role of GH in directly increasing MITF and MITF-regulated genes.

We further queried the changes in protein level expression of MITF and a few of its target genes in these melanoma cells, following drug treatment, GH treatment, as well as GHR blockade. In SK-MEL-30 cells, GH addition increased MITF expression by 1.8-fold (Figure 3A). Further, while doxorubicin addition also increased MITF protein levels by more than two-fold, addition of GH+doxorubicin caused an almost three-fold increase (Figure 3A). GHRKD strongly suppressed the upregulation of MITF in all conditions in SK-MEL-30 (Figure 3A), further validating the effect of GH in upregulating MITF expression. Doxorubicin treatment in presence of GH significantly raised expression level of MET as well as melanosomal marker PMEL, wherein a 1.5-fold downregulation of the effect was observed following GHRKD (Figure 3A). In the amelanotic melanoma SK-MEL-28, MITF protein levels were significantly elevated in presence of excess GH as well as doxorubicin but were again strongly suppressed following GHRKD (Figure 3B). Therefore, with the validation that GH upregulates and GHR-blockade downregulates MITF and MITF-target gene expression, we next wanted to examine if the MITF-regulated cellular activities were in turn activated by GH and suppressed by GHR blockade.

It is known that human GH has prolactin (PRL) activity. Therefore, in order to verify if PRL receptor (PRLR) activation affects MITF or MITF target gene expression, we analyzed the level of GHR and PRLR expression in all four human cell lines and observed that in RT-qPCR, the GHR Ct values were significantly higher than PRLR Ct values, reflecting a 10–30-fold higher expression of GHR compared to PRLR in these melanoma cell lines (Appendix A), similar to earlier reports [20]. Further, we treated SK-MEL-28 and MDA-MB-435 cells with 0 and 50 ng/mL recombinant human PRL, in absence and presence of doxorubicin for 24 h. Although we observed a mild increase in either PRL or PRLR in the human melanoma cell lines due to doxorubicin, we did not observe any upregulation of MITF or MITF targets due to PRL addition (Appendix A).

### 2.3. GH Upregulates MITF-Regulated Melanogenic Processes Following Drug Treatment in Human Melanoma Cells

MITF is the central regulator of melanocyte and melanoma development and is known to confer survival advantage to melanoma tumors, especially under therapeutic challenges by driving multiple processes, including melanogenesis [42,46]. In the previous sections, we observed distinct upregulation of multiple melanogenesis mediators and melanosomal proteins following the GH-induced increase in MITF expression. In order to determine if our observations result in an upregulated MITF activity, we chose to assay the tyrosinase activity as well as the melanin production rate in GH treated or GHR-blocked melanoma cells, in presence of multiple classes of anticancer drugs. Melanotan, an a-MSH analog, was used at 100 nM as a positive control to elicit melanin production in the melanotic melanoma cell line SK-MEL-30. In addition, SK-MEL-30 cells in triplicates in six-well plates were treated with chemotherapy -doxorubicin (1 μM); tyrosine kinase inhibitors—linsitinib (5 μM), crizotinib (10 μM), sorafenib (5 μM); and mutant BRAF targeted therapeutics—vemurafenib (2 μM) and dabrafenib (3 μM), for 72 h. At the end of the treatments, total melanin was quantified and normalized by cell number. We observed that chemotherapy doxorubicin elicited a modest 1.5-fold increase in melanogenesis at both 50 and 100 ng/mL GH, while GHRKD in presence of GH significantly suppressed the melanin increase (Figure 4A). Although dabrafenib caused significantly higher melanogenesis, which was again suppressed by GHRKD (Figure 4B), the highly efficacious V600E-BRAF-inhibitor vemurafenib did not cause any significant variation in total melanin production in SK-MEL-30. Among the tyrosine-kinase inhibitors, both linsitinib (IGF1R-inhibitor) and sorafenib (pan-tyrosine kinase inhibitor) significantly upregulated melanin production, while GH treatment further enhanced the effects (Figure 4C). GHRKD suppressed the GH-induced increase in melanogenesis in both cases (Figure 4C). Interestingly, the MET-inhibitor crizotinib did not elicit any significant change in the total melanin content in SK-MEL-30 cells (Figure 4C).

In addition to the above, we performed melanin quantitation for two other human melanoma cell lines, MDA-MB-435 and MALME-3M, both harboring a BRAF V600E mutation unlike SK-MEL-30, which has a wild-type BRAF. The 48 h treatments with doxorubicin, cisplatin, as well as the V600E-BRAF-inhibitor vemurafenib all increased the melanin production significantly in both MALME-3M (Figure 4D) and MDA-MB-435 (Figure 4E). Treatment with 50 ng/mL GH significantly enhanced the melanogenesis effects in both cell lines, especially for vemurafenib treatment (Figure 4D,E). An siRNA-mediated GHR knockdown either in presence or in absence of GH markedly suppressed the melanin production induced by drug treatment in both MALME-3M (Figure 4D) and MDA-MB-435 (Figure 4E). Melanin quantitation in mouse melanoma B16F10 cells also showed significant upregulation in all three drug treatments, while treatment with 50 ng/mL bovine GH enhanced the effects further (Figure 4F).

After doxorubicin treatment, we observed a marked increase in the tyrosinase activity in MDA-MB-435 and MALME-3M cell lines, wherein 50 ng/mL GH treatment further increased the effects (Figure 5A,B). In both SK-MEL-28 and SK-MEL-30 cell lines, doxorubicin treatment induced significantly higher tyrosinase activity only in presence of GH (Figure 5C,D). In all four cell lines, a consistent suppression of GH+doxorubicin-induced tyrosinase activity was observed following siRNA-mediated GHRKD (Figure 5A–D). Therefore, GH does appear to influence MITF-regulated melanogenesis pathways in human melanoma cells, albeit in a cell-line specific and drug-specific manner, which could be a critical determinant in the choice of combination therapies with GHR-antagonism as an approach to combat cancer drug resistance.

In the previous sections, we observed an increase in PGC1A RNA levels following GH treatment. In melanoma, PGC1A is a direct target of MITF [43,47,48] and is known to upregulate mitochondrial biogenesis in melanoma. In our investigations, PGC1A was upregulated by chemotherapy or GH alone, as well as by GH+doxorubicin treatment. Therefore, using Seahorse, we evaluated the extracellular acidification rate (ECAR) which indicates glycolysis rate in glycolysis-dependent cancer cells, including melanoma. Increasing doses of GH (0, 50, 200 ng/mL) increased ECAR in SK-MEL-28 cells in a dose-dependent manner (Appendix A). However, following 48 h doxorubicin treatment, the GH-induced ECAR was completely suppressed (Appendix A) despite increased PGC1A expression (Appendix A). This observation was in agreement with reports that therapeutic intervention in BRAF-mutant melanoma induces a PGC1A-driven shift from glycolysis to increased oxidative phosphorylation for improved reactive oxygen species (ROS) detoxification, thus conferring chemoresistance to the tumor cells [43,47,48].

### 2.4. Mouse Tumor Xenografts Show Increased MITF and MITF Target Gene Expression in bGH vs. WT Mice

Cells cultured in a monolayer often have limited MITF expression compared to in vivo tumors [49]. Therefore, in order to validate our in vitro observations, we obtained tumor samples from mouse B16F10 melanoma xenografts in immunocompetent C57BL/6J mice transgenic for bovine GH (bGH mouse) vs. the same in their wild-type (WT) littermates. Separate samples (*n* = 6) were obtained for male and female mice. Although the tumors did not vary markedly in growth and size, they did present a markedly different gene expression profile for several GH-regulated genes. In our RNA expression analyses, the tumors from bGH transgenic mice showed a nonsignificant trend towards an increase in MITF levels compared to WT, in both sexes (Figure 6A,B). In addition, the male bGH mouse tumors had significantly higher expression of MITF-targets, PMEL, DCT/TYRP2, MLANA, and BCL2 (Figure 6A), while the female mice also had significantly higher expression of PMEL and TYR (Figure 6B). Further, at the protein level, the male bGH mouse tumors had markedly elevated MITF and TYRP1 expression (Figure 6C,D), while the female mice tumors had markedly elevated PMEL and MET levels (Figure 6C,E). Interestingly, both male and female bGH mouse tumors had a higher endogenous GH level compared to WT mice (Appendix A). Therefore, in vivo, conditions of GH excess appear to elicit an increased expression level of MITF as well as MITF target genes more distinctly in the male mice, thus validating our in vitro observations. Incidentally, all the four cell lines used in the in vitro study are from male patients as indicated in the Methods section. In fact, in silico analysis of melanoma patient cohorts from the Cancer Genome Atlas (TCGA) database also showed a higher number of deaths in male patients with high GHR (above mean GHR expression) compared to low GHR (below mean GHR expression) when segregated by tumor stage (Appendix A). Thus, GH action appears to have sex specificity in the severity of its effect in melanoma, which globally has more than double the number of adult male patients than females [50,51].

### 2.5. GHR and MITF Strongly Correlate in Normal Human Skin as Well as Human Cancer Samples

In order to further validate our in vitro and in vivo observations, we analyzed the gene expression levels of GHR, MITF, and MITF targets in the NCI60 (National Cancer Institute) and CCLE (Cancer Cell Line Encyclopedia) datasets. In both cases, we observed a strong association and clustering of samples with high GHR and high MITF levels. This was in a high degree of agreement with the elegant study by Vazquez et al., where in the GSE7553 dataset for RNA expression of primary and metastatic melanoma patients grouped as per high and low PGC1A levels, GHR was the 10th and MITF was the 19th most differentially upregulated gene in the PGC1A-high cluster [47]. In our present analyses, we analyzed GHR, MITF, and 48 MITF target genes, including regulators of melanogenesis, in the NCI60 dataset (Figure 7A). We observed a strong clustering of all 50 genes, showing a robust association of MITF and GH action in human cancers. We observed a similar clustering in the CCLE dataset for the same 50 genes (Appendix A). Further, in analyses of the GTEx (Genotype Tissue Expression) human post-mortem dataset containing samples from 53 non-diseased tissue sites for nearly 1000 individuals, we observed a highly significant correlation between GHR and MITF expression in sun-exposed (lower-leg; Pearson coefficient = 0.19, *p* = 0.00071) and not sun-exposed (suprapubic; Pearson coefficient = 0.25, *p* = 0.00011) normal human skin (Figure 7B). Additionally, we queried the TCGA melanoma pan-cancer dataset and segregated the patients as GHR-high (above mean GHR FPKM value; *n* = 117) and GHR-low (below mean GHR FPKM value; n = 354) to look at MITF and MITF targets as well as ABC-transporter expression levels in male and female patients. Both sexes displayed an elevated level of multiple ABC-transporters as well as MITF-target genes (Appendix A).

### 2.6. GH-Regulated MITF and MITF Target Gene Regulation Proceeds via JAK2-STAT5 and SRC Regulated Pathways

Previously, we and others have described the spectrum of intracellular signaling cascades activated by GH binding to its cognate GHR dimer in human melanoma as well as in other human cancers [20,52,53,54]. Here, to elucidate which principal oncogenic signaling mechanism is involved downstream of GHR towards increasing MITF expression and activity, we performed a pathway inhibition analysis using JAK2-inhibitor-V (Calbiochem, Millipore Sigma, Burlington, MA, USA), STAT5-inhibitor (C16H11N3O3; CAS285986-31-4), SRC-inhibitor (saracatinib), or ERK-inhibitor (SCH-772984). Human melanoma cells MDA-MB-435 were incubated for 6 h with inhibitors of JAK2, STAT5, SRC, or ERK1/2 (p44/42-MAPK) at concentrations where they do not reduce cell viability by more than 20% in 24 h (Appendix A). After 6 h incubation with pathway inhibitors, 5 ng/mL GH +/−doxorubicin was added and RNA was collected after 16–18 h and analyzed. First, we queried the variations in the endogenous GH gene to estimate the effects of inhibitor treatments as we expected upregulation in GH production if the GH activated pathways are blocked. We observed massive increases in endogenous GH production (Figure 8), which was upregulated by >17-fold, >117-fold, >31-fold, and >18-fold by JAK2, STAT5, SRC, and ERK inhibition, respectively, in absence of any drug. Doxorubicin-treated samples showed >112-fold, >97-fold, >30-fold, and >15-fold following JAK2, STAT5, SRC, and ERK inhibition, respectively (Figure 8A). This did indicate that endogenous GH and the JAK2, STAT5, and SRC signaling pathways were critical for melanoma survival. We then analyzed changes in expression of MITF and MITF target genes PGC1A, PMEL, MET, and TYRP1 following pathway inhibition. GH-induced MITF upregulation was suppressed two-fold by STAT5-inhibition and >three-fold by SRC-inhibition, but not significantly by JAK2 or ERK inhibition. However, the MITF upregulation in presence of GH+doxorubicin was maximally suppressed by STAT5 and SRC inhibitions (>four-fold for both) again, but also to 1.7-fold by ERK-inhibition (Figure 8). Thus, there appears to be an involvement of the GH-regulated MAPK pathway in MITF regulation in presence of chemotherapy, which does agree to previous reports [55,56]. MITF-targets TYRP1, PMEL, and PGC1A—all showed marked suppression of GH+doxorubicin-induced upregulation in expression with JAK2, STAT5, and SRC inhibitor treatments, with SRC inhibition showing the maximum reductions in all cases (Figure 8). MET expression was suppressed by both JAK2 and STAT5 inhibitions but not by SRC or ERK inhibition (Figure 8). Interestingly, ERK-inhibition alone markedly upregulated the expression levels of both MITF, PGC1A, TYRP1, PMEL, and MET in the presence of GH. This effect surprisingly and accurately recapitulates earlier reports of MITF- and PGC1A-mediated pathways of chemoresistance following treatment with inhibitors of the MAPK pathway (such as mutant BRAF inhibitors) [43,47,48]. This observed effect is also of critical relevance to the approach of combining GHR antagonism with BRAF-inhibition therapy, where GHR-antagonism can effectively abrogate the GH-induced resistance to BRAF-inhibitors. Similar suppression of MITF and MITF target genes were also observed in SK-MEL-28 (Appendix A) and SK-MEL-30 (Appendix A) due to JAK2, STAT5, and SRC inhibitions. Therefore, our results implicate the JAK2-STAT5 as well as the SRC pathway as the principal molecular mediators of GH-mediated upregulation of MITF expression and activity in melanoma.

## 3. Discussion

MITF is a ‘lineage-addiction’ or ‘lineage survival’ oncogene in melanoma, amplified in as much as 10% of primary tumors and >20% of metastatic tumors [34,35]. There is a marked correlation between MITF amplification in tumors and tumoral therapy resistance [42]. Single cell qPCR of melanoma patient primary tumors and distant metastases have revealed an MITF signature in intra- and intertumoral heterogeneity with coexpression of three distinct clusters of cells—MITF-high, MITF-low, and a subset expressing markers of both [57]. This tumoral heterogeneity renders a high level of plasticity to melanoma tumors to switch between phenotypes of proliferation to invasion to stemness, along a high to low MITF gradient [56], thereby abetting the characteristic extreme therapy refractoriness of melanoma. Adaptive responses to the dynamic tumor microenvironment are also largely regulated by MITF [58], wherein MITF along with 74 MITF target genes, was found to be the central transcriptional regulator in melanospheres—an anchorage independent cluster of circulating melanoma cells with high self-renewal and cancer stem-cell-like properties [49]. Immunohistochemical analyses of melanoma patients with differential response to DNA-alkylating chemotherapy agents dacarbazine or temozolomide identified high MITF and melanosomal protein levels in the nonresponsive group [59]. Conversely, melanoma with dysfunctional melanosomal and melanogenic machinery have increased cisplatin sensitivity [27]. One of the most critical effects of the melanoma oncogene MITF is therefore in therapy resistance—a phenomenon in cancer cells which also strongly implicates GH action [14], especially in metastatic (stage IV) melanoma [21], which also displays the highest GHR expression in the NCI60 panel [6] of 60 human tumor cell lines of eleven different cancer types. Although the exogenous growth factors bFGF, EGF, and HGF have been found to not affect V600E-BRAF melanoma viability or response to BRAF-inhibitors [60], we have previously reported that GHR inhibition sensitizes human melanoma to the BRAF inhibitor vemurafenib by suppressing ABC-transporter-mediated clearance of vemurafenib [21]. The complex regulation of MITF expression in the highly heterogenous tumor population is affected by multiple upstream signaling pathways, genetic and epigenetic alterations, as well as by the melanoma microenvironment. In the current study, we present in vitro, in vivo, and in silico evidence describing a GH-mediated upregulation of MITF expression and activity, with implications in melanoma drug resistance.

We observed an increase in GH and MITF levels in melanoma cells following treatment with anticancer drugs. Previous reports have also found GHR and MITF as two of the top-20 upregulated genes in the PGC1A-high cohort of GSE7553 dataset of primary and metastatic melanoma patient gene expression [47,61]. The PKA-activated cAMP response element binding protein (CREB) drives the expression of both GH [62] and MITF [56,63], whereas GH in turn also mediates transactivation of CREB by rapid phosphorylation at Ser133 via p44/42-MAPK (ERK1/2) [64,65], and can thus directly regulate MITF expression. This is of importance particularly in the context of chemotherapy, where we observed that MAPK blockade does in turn block MITF expression in MDA-MB-435 cells, but not in the absence of chemotherapy, where STAT5 and SRC are the primary mediators of GH regulated MITF upregulation. Moreover, both phosphorylated and unphosphorylated ERK1 and ERK2 have been shown to directly bind to MITF with potential to regulate MITF target gene expression [55]. This might partly explain our observation that ERK1/2 inhibitors significantly increased GH-mediated MITF and MITF targets’ (TYRP1, PGC1A, PMEL, and MET) expression in MDA-MB-435 cells, in absence of chemotherapy. Our data therefore suggest a bimodal transcriptional regulation of MITF expression by GH based on the therapeutic context. This allows for a deeper understanding of the enigmatic ‘rheostat-model’ of MITF regulation in melanocytes and melanoma and can be validated by future studies. Additionally, the melanin-concentrating hormone (MCH) as well as neuropeptide-E1 (NE1), reported to strongly induce GH secretion from cultured rodent and human pituitary cells [66], also binds to mouse B16 melanoma cells [67] and increases melanin and cAMP production [68]. Further, melanoma GH production could be a function of the DNA-damaging effects induced by chemotherapy, as was described in colonic neoplasms by Chesnokova et al. [16]. 

Our in vitro, in vivo, and in silico studies described a distinct upregulation of multiple MITF-regulated pro-survival and therapy-refractory pathways and its mediators due to direct GH treatment, especially in the presence of anticancer drugs. MITF has several targets [44] and we queried a selected handful of them in this study, based on their role in cancer therapy resistance. The known MITF targets that were upregulated by GH treatment and suppressed by GHRKD, and further corroborated by in vivo or in silico studies included BCL2—an antiapoptotic protein, CDKN1A—a cell cycle inhibitor as well as MITF expression promoter, BRCA1—a DNA repair protein, and MET—a proto-oncogene in multiple cancers. MITF-mediated MET upregulation is known to elicit antiapoptotic effects in melanocytes and melanoma [36], while our earlier reports did show a GH-mediated increase in MET levels in human melanoma [20]. Our current study identifies MITF as a partial mediator of GH-regulated MET upregulation and strongly implicates GH as an upstream effector in promoting MITF-regulated antiapoptotic and pro-survival effect in melanoma. Additionally, we also detected MITF targets involved in combating oxidative stress—PGC1A and HIF1A are significantly modulated by GH action. PGC1A and MITF are a notorious pair implicated in overcoming melanoma’s glycolytic dependence [48,69] and in development of rapid resistance to BRAF-inhibitor therapy [43,47], a cornerstone in recent melanoma therapeutics [70], and are currently in clinical trials as an adjuvant in melanoma immunotherapy [71]. Additionally, PGC1A upregulation is a key driver of metastasis in multiple cancer types and is enhanced in circulating tumor cells compared to primary tumors [48]. Our study reaffirms and provides an explanation to the coexpression of GHR, MITF, and PGC1A in primary and metastatic melanoma patients [47,61]. This is again a critical finding with therapeutic value in combining GHR antagonists with BRAF-inhibitors as a means of over-riding MITF- and PGC1A-mediated BRAF-inhibitor resistance and needs to be further investigated with appropriate in vivo and clinically relevant models. Also, the direct effects of GH action on PGC1A is unknown and could be an important aspect in explaining some of the survival advantages bestowed by congenital GH resistance (Laron Syndrome).

Lastly, one of the highlights of our current study is identification of the effect of GH action in melanoma in promoting MITF-mediated melanogenesis in melanosomes. In malignant melanoma, drug sequestration in melanosomes has been reported to contribute to chemoresistance to cisplatin [26], similar to the effects of vesicular drug sequestration conferring cancer drug resistance [24,72]. ABC-transporters, which we and others have previously described as targets of GH–GHR action in melanoma [21] and breast cancer [22], are also implicated in active sequestration of drugs in melanosomes [28]. We observed GH-induced as well as GH +chemotherapy-induced activation of the melanogenesis process in melanoma, as indicated by our melanin quantitation and tyrosinase activity assays. Additionally, we reported a GH-induced upregulation and GHRKD-induced suppression of MITF-regulated melanogenesis pathway genes—TYR, TYRP1, TYRP2/DCT—as well as melanosomal markers PMEL (gp100) and MLANA, in multiple melanoma cell lines, xenograft models, and in silico analyses. Johansson et al. have reported that blocking PMEL expression sensitized melanoma cells to paclitaxel and cisplatin treatments [59], while mutational silencing of melanosomal proteins also attenuates multidrug resistance in metastatic melanoma [27]. Melanogenesis is an intrinsic protective mechanism against UV rays and reactive oxygen species (ROS) in normal melanocytes and prevents UV- and ROS-induced melanocyte transformation. In melanoma, this unique phenomenon is harnessed as an impediment for radiation therapy [73]. Importantly, autocrine GH promotes resistance to radiation therapy in breast and endometrial cancers [17]. Therefore, targeting GH action could re-sensitize melanoma tumors to both chemotherapy and radiation therapy.

## 4. Materials and Methods

### 4.1. Cell Culture and Treatments

Human melanoma cell lines MALME-3M (male/43 year), SK-MEL-28 (male/51 year), and MDA-MB-435 (male/33 year) were obtained from ATCC (Manassas, VA, USA), while SK-MEL-30 (male/67 year) cell-line was acquired from Creative Bioarray (Shirley, NY, USA) with the help of Annie Zhao. Cells were maintained in IMDM, EMEM, RPMI-1640, and EMEM culture media, respectively, supplemented with 10% fetal bovine serum (FBS) and 1X-penicillin–streptomycin—all purchased from ATCC. Mouse melanoma cells B16F10 (male) were also obtained from ATCC and maintained in DMEM media supplemented with 10% FBS and 1X-penicillin–streptomycin. Treatment with indicated concentrations of recombinant human GH (Antibodies Online, Limerick, PA, USA) for indicated time-points or anticancer drugs were performed as described. Anticancer compounds doxorubicin (chemotherapy), cisplatin (chemotherapy), crizotinib (MET-inhibitor), linsitinib (IGF1R-inhibitor), sorafenib (tyrosine-kinase inhibitor), cabozantinib (MET+VEGFR2-inhibitor), dabrafenib (V600E-BRAF-inhibitor), and vemurafenib (V600E-BRAF-inhibitor) were obtained from Selleckchem (Houston, TX, USA).

### 4.2. Mouse Melanoma Xenografts

Five million B16F10 mouse melanoma cells were injected subcutaneously to the flanks of 12-week old C57BL/6J wild-type (WT) or bovine-GH transgenic (bGH) mice, and the size of perpendicular tumor diameters was measured using a digital caliper. The tumor growth was allowed until the tumors grew to 2 cm^2^ (~3 weeks), at which point the animals were sacrificed and the tumors were surgically removed and frozen at −80 °C. All animal handling procedures were performed in accordance with policies under Ohio University IACUC policy (#16-H-016). Protein and reverse transcribed RNA (cDNA) samples from mouse tumors were analyzed for expressions at RNA and protein of selected targets.

### 4.3. RNA Extraction and RT-qPCR

RNA was extracted and RT-qPCR performed as previously described [20,21]. Briefly, following treatments, total RNA was extracted using IBI Scientific (Dubuque, IA, USA) Total RNA extraction kit, following manufacturer’s protocol. From extracted RNA, mRNA levels were quantified using oligo-dT reverse transcription followed by quantitative PCR using Applied Biosystems (ThermoFisher, Waltham, MA, USA) reagents following manufacturer’s protocol. Primer sequences are provided in Appendix A.

### 4.4. Protein Extraction and Western Blot

Protein extraction and Western Blot were performed as described before [20,21]. Briefly, protein extraction was done using 1X RIPA buffer (SIGMA-Aldrich, St. Louis, MO, USA) containing 1X phosphatase-protease inhibitor cocktail (Cell Signaling Technology, Danvers, MA, USA). Protein concentration was estimated using Bradford assay and 80 μg of protein were loaded onto 4–16% gradient denaturing gels, transferred to PVDF membranes, blocked with 5% BSA solution in 1X TBS-T and probed using target-specific primary and secondary antibodies. List of antibodies and antibody sources are provided in the Appendix A.

### 4.5. Melanogenesis (Melanin Quantitation) Assay

The melanogenesis assay was performed as described before [74]. Briefly, melanoma cells were plated at 500,000 cells/well and treated with MSH-analog melanotan or anticancer compounds at mentioned concentrations for 72 h, +/−GH, +/−siRNA-mediated GHRKD. After 72 h cells were trypsinized, counted, pelleted by centrifugation for 10 min at 10,000× *g*, and the pellets were dissolved in 300 μL 2N NaOH for 30 min at 60 °C. The absorbance of the sample was measured at 450 nm using a microplate reader. Final absorbance values were normalized by cell number. Data represents mean +/−SD. Student’s *t*-test was performed using MS-EXCEL to identify significant differences at *p* < 0.05.

### 4.6. Tyrosinase Assay

Tyrosinase activity of melanoma cell lysates (treated as per mentioned in text with +/−GH, +/− si-RNA-mediated GHRKD, +/−anticancer compounds using a tyrosinase activity assay kit (cat#K742) from BioVision Inc. (Milpitas, CA, USA) following manufacturer’s protocol. Twenty-five micrograms of melanoma cell lysate were used per assay.

### 4.7. siRNA-Mediated GHR Knockdown

GHR knockdown using GHR-specific siRNA was performed as previously described [20,21]. Briefly, three different sequences of GHR-specific siRNA from Origene (Rockville, MD, USA) were used and siRNA-B at 25 nM provided maximum GHR knockdown efficiency in all cell lines tested. Transfection was performed using siLentFect (Biorad, Hercules, CA, USA) reagent as per manufacturer’s protocol.

### 4.8. Bioinformatic Analysis

Multiple datasets from the cBioPortal for Cancer Genomics were analyzed. The Z-scores (deviation from mean value) for RNA expression of GHR, MITF, and MITF targets were analyzed to generate RNA expression heatmaps—(i) National Cancer Institute-60 Cell Lines (NCI—67 samples, and (ii) Cancer Cell Line Encyclopedia (CCLE—967 samples). The cBioPortal for Cancer Genomics (http://www.cbioportal.org/) is an open-access resource for visualization and analysis of >5000 tumor samples from 105 cancer studies. Pearson’s correlation analysis of GHR and MITF expression in normal human skin (sun-exposed and not sun-exposed) from the GTEx (Genotype Tissue Expression) dataset was performed using the open-access GEPIA platform (Gene expression profiling interactive analysis) [75].

### 4.9. Statistical Methods

Parametric and nonparametric statistical analyses for comparing RNA levels were done using R software (ver3.0.2). For RT-qPCR analysis of RNA, the levels were first normalized against two reference genes (GAPDH and beta-actin) and the 2^−ΔΔCt^ values were compared by Wilcoxon signed rank test for significance. A *p*-value less than 0.05 was considered as significant. The densitometry analyses, melanogenesis assay, tyrosinase assay, and resazurin based assays, were compared by a paired Student’s *t*-test and ANOVA was performed (using GraphPad Prism software version7.04) to compare for significance (*p* < 0.05 is considered significant).

## 5. Conclusions

In conclusion, the present study provides a valuable mechanistic insight into therapy resistance regulation by GH and GHR in human cancers. Keeping in view the cumulative information from earlier research in the field of GH and cancer [14,76] as well as our current findings, combined GHR antagonism appears to offer a method of making radiotherapy, chemotherapy, as well as specific targeted therapies more efficacious not only in melanoma, but also in other solid tumors which overexpress the GHR. This might be a critical treatment modality for millions of cancer patients worldwide. Appropriately designed in vivo and clinical studies will be invaluable and essential to bring the approach of combined GHR antagonism to overcome tumoral resistance in existing melanoma therapies from bench to bedside.

## Figures and Tables

**Figure 1 cancers-11-01352-f001:**
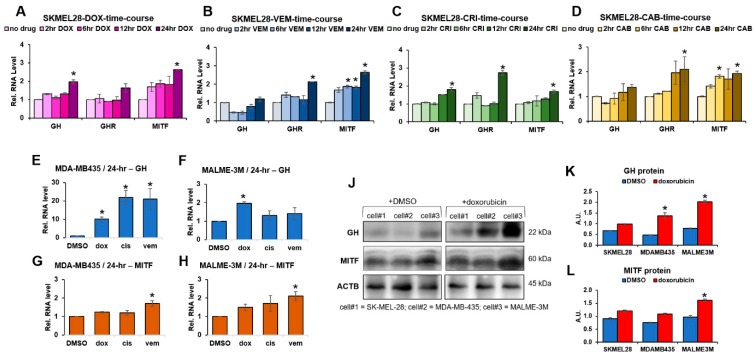
Changes in gene expression following drug-treatment in human melanoma cells: (**A**–**D**) Human melanoma cells SK-MEL-28 were treated with anticancer drugs and RNA expression was compared with respective untreated controls, at 2, 6, 12, and 24 h timepoints by RT-qPCR with corresponding prevalidated primers (sequence in Appendix A) for target genes. Changes in growth hormone (GH), growth hormone receptor (GHR), and melanocyte-inducing transcription factor (MITF) and ABC-transporters (Appendix A) with time due to doxorubicin (**A**), vemurafenib (**B**), crizotinib (**C**), and cabozantinib (**D**) treatments reflected concomitant changes in MITF, GH–GHR axis (**A**–**D**) as well as in ABC-transporters (Appendix A). (**E**–**H**) Changes in RNA expressions of GH (**E**,**F**) and MITF (**G**,**H**) RNA expressions at 24 h treatment with doxorubicin (dox), or cisplatin (cis), or vemurafenib (vem) treatments on MDA-MB-435 (**E**,**G**) and MALME-3M (**F**,**H**) cells. RNA expressions were quantified by RT-qPCR and normalized against expression of TUBB5 and ACTB as reference genes (*, *p* < 0.05, Wilcoxon sign rank test, *n* = 3). (**J**–**L**) In SK-MEL-28, MDA-MB-435, and MALME-3M human melanoma cells, treatment with doxorubicin for 48 h increased the protein levels of GH and MITF as seen by western-blot (**J**) and subsequent quantification using ImageJ (NIH) (**K**,**L**) and expressions were normalized against expression of ACTB (β-actin) (*, *p* < 0.05, Students *t* test, *n* = 3).

**Figure 2 cancers-11-01352-f002:**
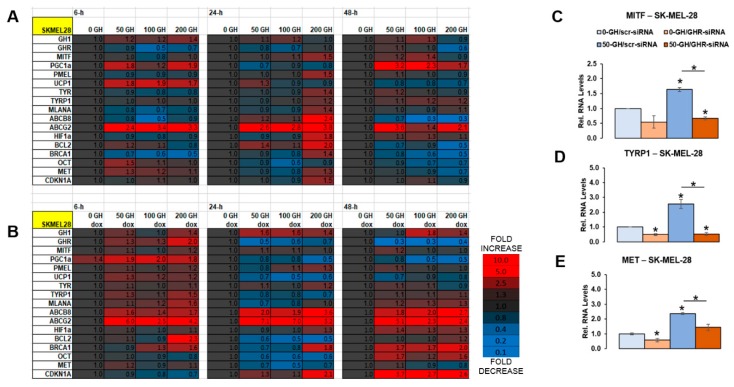
GH treatment directly upregulates MITF and MITF target RNA expression in human melanoma cells: (**A**,**B**) Human melanotic melanoma cells SK-MEL-28 were treated with increasing doses (0, 50, 100, 200 ng/mL) of recombinant human growth hormone (GH) and heatmap showing changes in RNA expressions at 6, 24, and 48 h timepoints were analyzed for GH, GHR, MITF, and a number of MITF targets, as well as the ABC-transporters ABCB5 and ABCG2 (**A**); An identical experiment was performed in the presence of 200 nM doxorubicin (**B**). Numbers inside boxes indicate fold-change in gene expression compared to GH untreated control. Similar set of experiments for amelanotic melanoma cells SK-MEL-28 is shown in Appendix A. (**C**–**E**) SK-MEL-28 cells were further transfected either with scramble (scr) of GHR-targeted (GHR) siRNA for gene knockdown in presence or absence of 50 ng/mL GH treatment and MITF (**C**), TYRP1 (**D**), and MET (**E**) RNA expressions were analyzed. RNA expressions were quantified by RT-qPCR and normalized against expression of TUBB5 and ACTB as reference genes (*, *p* < 0.05, Wilcoxon sign rank test, *n* = 3).

**Figure 3 cancers-11-01352-f003:**
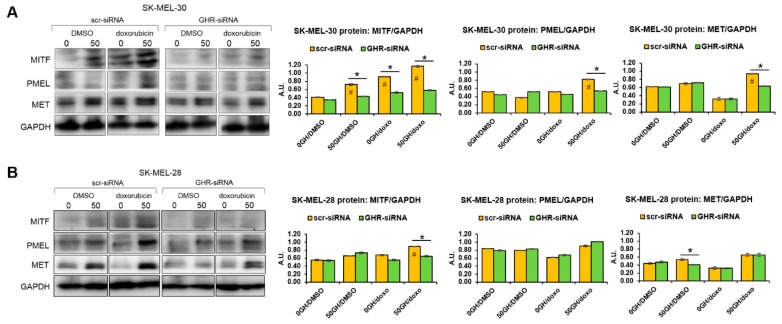
GH treatment directly upregulates MITF and MITF target proteins’ expression in human melanoma cells: (**A**,**B**) Human melanotic melanoma cells SK-MEL-30 (**A**) and SK-MEL-28 (**B**) were treated with/without 50 ng/mL recombinant human growth hormone in presence/absence of doxorubicin and siRNA-mediated GHR knockdown (GHRKD). After 48 h, protein was extracted and western-blot was performed for MITF, PMEL (PMEL17/gp100), MET, and GAPDH followed by densitometry analysis performed using ImageJ (NIH) and expression were normalized against expression of GAPDH as loading control (#,*, *p* < 0.05, Students *t* test, *n* = 3; *#* indicates comparison against untreated (-GH, -dox, -GHRKD) control).

**Figure 4 cancers-11-01352-f004:**
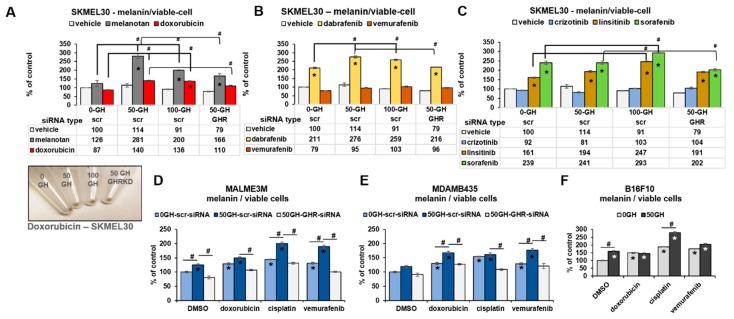
GH enhances MITF regulated melanogenesis activity in human melanoma cells. Changes in amounts of melanin formed inside SK-MEL-30 cells following treatment with different treatments for 72 h in presence of 0, 50, 100 ng/mL recombinant human growth hormone as well as siRNA-mediated GHR knockdown (GHRKD) in presence of 50 ng/mL GH was analyzed. (**A**) alpha-MSH analog melanotan (100 nM), doxorubicin (1 μM); (**B**) mutant BRAF targeted therapeutics—vemurafenib (2 μM), and dabrafenib (3 μM); or (**C**) tyrosine kinase inhibitors—linsitinib (5uM), crizotinib (10 μM), sorafenib (5 μM) were used. Similar treatments with doxorubicin, cisplatin, and vemurafenib was done with BRAF-mutant cell-lines MALME-3M (**D**) and MDA-MB-435 (**E**) as well as the mouse melanoma cell line B16F10 (**F**). Cell pellets were dissolved by heating in NaOH and absorbance was measured at 450 nm. Reads were normalized by cell counts at the end of 72 h treatments. Inset shows cell pellets from doxorubicin treatment. Data represents mean +/− SD. Student’s *t*-test was performed using MS-EXCEL to identify significant differences at *p* < 0.05. (#,*, *p* < 0.05, Students *t* test, *n* = 3; *** indicates comparison against untreated (-GH, -dox, -GHRKD) control).

**Figure 5 cancers-11-01352-f005:**
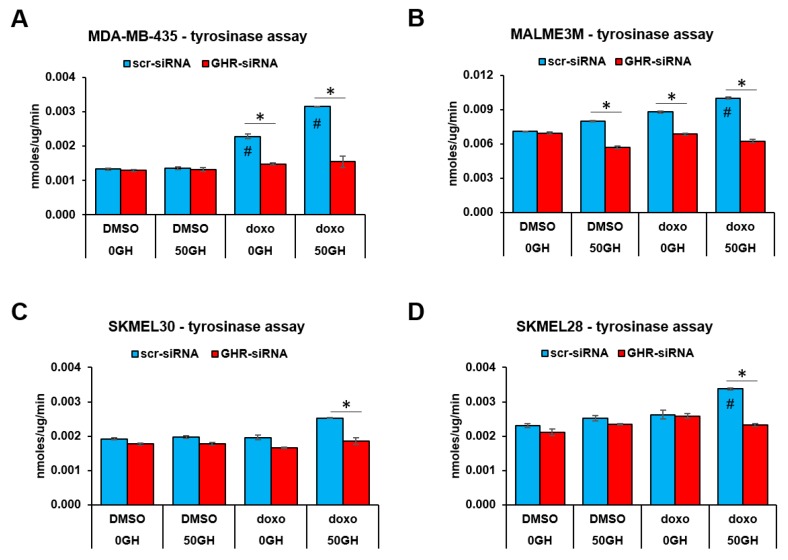
GH enhances MITF-regulated tyrosinase activity in human melanoma cells. Changes in tyrosinase activity in 25 μg cell lysates of (**A**) MDA-MB-435, (**B**) MALME-3m, (**C**) SK-MEL-30, and (**D**) SK-MEL-28 cells treated with/without 50 ng/mL recombinant human growth hormone in presence/absence of doxorubicin and siRNA-mediated GHRKD. After 48 h treatment, total protein was extracted and quantified by Bradford assay. Tyrosinase assay was performed using a commercial kit (BioVision) following manufacturer’s protocol (#,*, *p* < 0.05, Students *t* test, *n* = 3; *#* indicates comparison against untreated (-GH, -dox, -GHRKD) control).

**Figure 6 cancers-11-01352-f006:**
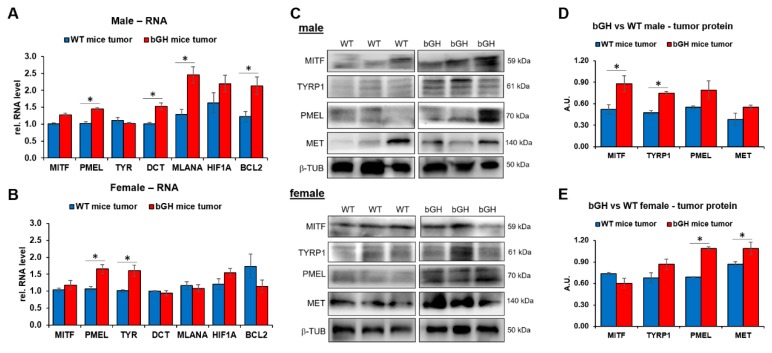
B16F10 mouse melanoma xenografts in bovine growth hormone transgenic (bGH) mice had elevated expression of MITF and MITF targets compared to the same in WT mice. Mouse melanoma B16F10 cells were xenografted in immunocompetent bovine growth hormone transgenic mice (bGH) as well as its wild-type littermates (WT) of both sexes (*n* = 6). Tumors grown for three weeks were extracted and levels of RNA and protein were analyzed by RT-qPCR and western-blot. Excess GH in bGH mice showed upregulation of multiple target RNA and protein levels compared to WT mice. (**A**,**B**) Changes in RNA levels of MITF and MITF-target genes in (**A**) male and (**B**) female bGH and WT mice. (**C**–**E**) Changes in protein levels of MITF and MITF-targets in male (**D**) and female (**E**) bGH and WT mice as seen by western-blot (**D**) and quantified by densitometry analysis performed using ImageJ (NIH) and expression were normalized against expression of TUBB5 as loading control (*, *p* < 0.05, Students *t* test, *n* = 3).

**Figure 7 cancers-11-01352-f007:**
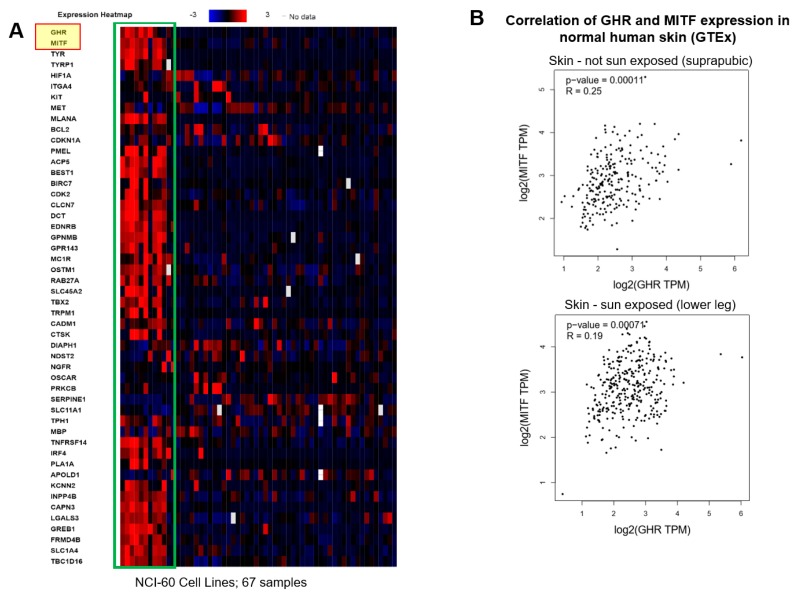
Bioinformatic analysis: GHR, MITF, and MITF target genes coexpress and strongly cluster in the human sample datasets. (**A**) Heatmap shows normalized RNA expression (FRKM) values of GHR, MITF, and 48 MITF target gene expression in all the samples of National Cancer Institute’s NCI-60 cancer cell panel. Green box indicates clustering of GHR, MITF, and MITF targets. Appendix A shows similar analyses for the CCLE (Cancer Cell Line Encyclopedia) dataset. (**B**) Correlation (Pearson’s) analysis of GHR and MITF RNA expression in the GTEx (Genotype Tissue Expression) human post-mortem dataset containing samples from 53 non-diseased tissue sites for nearly 1000 individuals using the GEPIA platform. A positive and highly significant correlation between GHR and MITF expression in sun-exposed (lower-leg; Pearson coefficient = 0.19, *p* = 0.00071) and not sun-exposed (suprapubic; Pearson coefficient = 0.25, *p* = 0.00011) normal human skin was observed.

**Figure 8 cancers-11-01352-f008:**
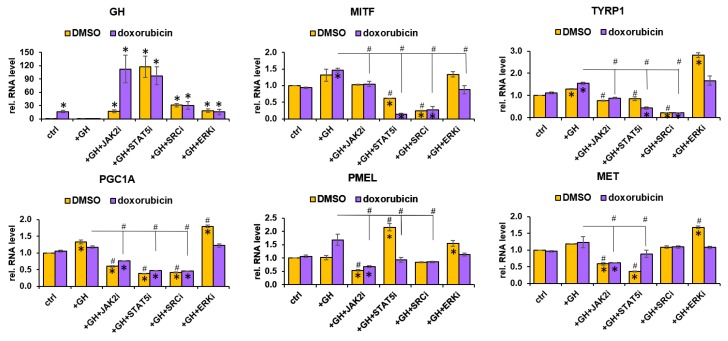
GH-regulated MITF and MITF target gene regulation proceeds via JAK2–STAT5- and SRC-regulated pathways: Human melanoma cell MDA-MB-435 (here), SK-MEL-28 (Appendix A) and SK-MEL-30 (Appendix A) were treated with/without doxorubicin in the presence of GH as well as different intracellular signaling pathway inhibitors. After 24 h treatment, RNA expression for target genes (GH, MITF, and MITF targets) was quantified by RT-qPCR and normalized against expression of TUBB5 and ACTB as reference genes (#,*, *p* < 0.05, Wilcoxon sign rank test, *n* = 3; * indicates comparison against corresponding -GH controls while # indicates comparison against corresponding +GH controls in DMSO and doxorubicin treated groups).

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
