# Peer review of "Growth Hormone Upregulates Melanocyte-Inducing Transcription Factor Expression and Activity via JAK2-STAT5 and SRC Signaling in GH Receptor-Positive Human Melanoma"

_cancers, 2019, doi:10.3390/cancers11091352_

Round 1
Reviewer 1 Report
A series of well-designed and superbly executed studies have been undertaken to address the growth hormone effects on melanocyte inducing transcription factor (MITF) expression by tumor cells. The paper is broadly well written.
Major concern
The authors used human GH which has prolactin activity. The manuscript would be greatly improved if the authors examined the effects of prolactin in a dose dependent manner.
Minor issues
Throughout - use of the word “expressions” not the correct “expression”. Figure 1 Please re-title adding clarity. Figure 3 Please use larger fonts Figure 6 legend. Please define bGH mice. I suggest bGH transgenic mice. I don’t like the term protein expression.
Reviewer 2 Report
This is a comprehensive survey of many melanoma cell lines that establishes a link between autocrine growth hormone-induced melanogenesis (via activated MITF) and drug resistance. The expression levels of a goodly number of downstream MITF targets were examined at the RNA and protein levels as well as a handful of ABC transporters involved in drug efflux in the absence or presence of GH and multiple kinds of ant-cancer drug treatments. They established that that not only is GH upregulated in melanoma cells, but addition of the drugs enhances expression of MITF, melanogenic factors and ABC transporters via Jak2-STAT5 and src signaling. Retrospective analysis of various public databases corroborated their findings. Together, this enormous amount of data provide an interesting and mostly compelling story.
The authors do show that the different efflux transporters show enhanced expression with the various GH/drug treatments. Does this upregulated expression correlate with enhanced drug efflux? Are the transporters expressed at the cell surface? It would be useful if the authors could comment on this.
Similarly, the authors provide evidence for enhanced melanogenesis and comment on the observation that the transporters mediate drug sequestration into melanosome in extreme drug resistant forms of melanoma. Can the authors show such activity in their studies? Can they comment on numbers of melanosomes in their treated cells or patient samples?
Reviewer 3 Report
The authors demonstrate the ability of GH and GHR to show how MITF mediates previously described published changes in gene expression (Ref #22). It seems in the earlier publication, much of the data were reported, the potential novelty here is the role of MITF. In the presented data, the role of GHR activating MITF is implied, but never shown directly and this is a major weakness. There are no experiments showing that suppression of MITF abrogates or diminishes the effects of GH on the gene expression profiles outlined.
Authors should outline the BRAF mutational status in each cell line used to better interpret the vemurafenib experiments. In the first figure, the authors characterize the effects of several cancer drugs on regulation of specific genes. Some reference back to the previous publication showing that the concentrations used are at IC50. It would be useful to have a negative control using a drug that has no effect on the cells to see if the effects on gene expression a sign of general cellular stress are just as opposed to a specific effect of melanoma chemotherapy. The authors chose to select different cell lines in Figure 1, making it difficult to follow the arguments. For example, Fig 1A-D use the SK-MEL-28 cell line, but E-H do not. Figure 2 also shows selective data with the SK-MEL-30 cell line. The cell line previously shown in Figure 1 (SK-MEL-28) are in the supplemental data. Yet a different cell line is in Fig 2C. Again, there is a switch of cell lines from the previous figure that make the argument difficult to follow. The authors suggest that GHR results in upregulation of MITF with subsequent increased transcription of several target genes. To demonstrate this, siRNA to GHR is used in Figure 3. However, there are no experiments demonstrating the downregulation of the target protein, GHR. While immunoblotting may be difficult, receptor binding sites could be shown. Further, the complementary experiment is not shown; there is no direct downregulation of MITF showing that it is necessary for GHR action. Figure 4 shows melanin production in one cell line. Since more than one melanotic cell line is described, additional data should be included. Figure 6 describes experiments in the B6F10 melanoma cell line. This cell line has never been discussed prior to this experiment. Again, switching of cell lines makes it difficult to follow the authors’ investigational thread. Some in vitro data should be included to bolster the findings from the in vivo experiments. As written, it seems that these data from another investigator’s experiment are tacked on the work. Further, these animals were engineered to express bGH, so the idea that autocrine GH has a role in melanoma biology is not supported by this mouse experiments. It seems a xenograft experiment using the cell lines in the previous aims would be more supportive of the role for autocrine GH. In Figure 7, human specimens are examined. Why wasn’t GH included in the analysis? Figure 8 shows effects of signal transduction pathways in the regulation of several genes. There are pharmacologic antagonists of these pathways and could be included in the analysis especially since the authors are making an argument for co-targeting of melanoma with drugs and GH antagonism.
Round 2
Reviewer 3 Report
The authors have made substantial revisions to the manuscript to address previous critiques. There are still areas of concern remaining.
Previous overall critique - In the presented data, the role of GHR activating MITF is implied, but never shown directly and this is a major weakness. The authors agree that this has not been shown. Since the heart of the paper is focused on GHR upregulation of MTF affecting melanoma cell biology, without showing this mechanism, the work implies direct cause-and-effect, but does not show this.
Previous critique 2 – The authors do not address how the concentrations of drugs were determined. There remain no negative controls.
Previous critique 4 – The downregulation of GH protein is still not shown. Data demonstrating steady state levels of GH mRNA are given, this may not correlate with functional protein levels.